# The dynamics of explore-exploit decisions suggest a threshold mechanism for reduced random exploration in older adults

Caroline E. Phelps [1,2]*, Alec E. Frisvold[1], Meghna Sreeram[1], Victoria D. Antoniou[1], Larissa E. Oliveira[1], Sierra R. Tooke[1], Yan Z. Lu[1], Anna C. Lemon[1], Joseph U. Fraire[1], Alexe G. Delval[1], Betsy K. Smith[1], Devynn H. Spangenberg[1], Madelien N. Mithelavage[1], Michelle N. Ngo [1], Elizabeth M. O. Keller[1], Liliana J. Issac[1], Sara A. Harader[1], Jack-Morgan Mizell[1], Siyu Wang [3], Waitsang Keung[1], Mark H. Sundman[1], Mary-Kathryn Franchetti[1], Ying-hui Chou[1,4], Gene E. Alexander[1,4,5,6,7], Robert C. Wilson[1,2,8]*

1 Department of Psychology, University of Arizona, Tucson, Arizona, United States of America, 2 Department of Psychology, Georgia Institute of Technology, Atlanta, Georgia, United States of America, 3 Laboratory of Neuropsychology, National Institute of Mental Health/National Institutes of Health, Bethesda, Maryland, United States of America, 4 McKnight Brain Research Foundation, University of Arizona, Tucson, Arizona, United States of America, 5 Neuroscience and Physiological Sciences Graduate Interdisciplinary Program, University of Arizona, Tucson, Arizona, United States of America, 6 Department of Psychiatry, University of Arizona, Tucson, Arizona, United States of America, 7 Arizona Alzheimer's Consortium, Phoenix, Arizona, United States of America, 8 Cognitive Science Program, University of Arizona, Tucson, Arizona, United States of America

* carolinephelps@gatech.edu (CEP); bob.wilson@gatech.edu (RCW)

## Abstract

When faced with a choice between exploring an unknown option vs exploiting an option they know well, older adults explore less and exploit more than younger adults. Recent work has suggested that one cause of this age difference in exploration is a reduction in the extent to which older adults use "random exploration" – exploration driven by behavioral variability. Here we investigate potential mechanisms for this age-related difference in random exploration through the lens of a drift diffusion model (DDM) of the explore-exploit choice. In this model, random exploration can be modulated by two mechanisms – the fidelity with which information about the choice is represented in the brain, the "signal-to-noise ratio" (SNR), and the amount of information required to make a decision, the "decision threshold." Reduced random exploration in aging could be caused either by an increase in signal-to-noise ratio or an increase in decision threshold in older adults. By fitting the DDM to choices and response times in a sample of healthy younger and older adults, we found that older adults had a lower SNR and a higher threshold than younger adults. This suggests that reduced random exploration in aging is driven by higher response thresholds in older adults, which may compensate for the reduced signal-to-noise ratio with which decision information is represented in the brain.

**Data availability statement:** All data and code can be found in an OSF repository Link: https://osf.io/upn8t/.

**Funding:** This work was supported by NIH grants P30 AG019610 Arizona Alzheimer's Consortium Pilot Study Program (PI: R.C.W., Co-I: Y.-H.C.), and R01 AG061888 (PI: R.C.W, Co-Is: Y.-H.C. and G.E.A.). The funders had no role in study design, data collection and analysis, decision to publish, or preparation of the manuscript.

**Competing interests:** The authors have declared that no competing interests exist.

## Author summary

The balance of deciding to explore the unknown, versus exploiting the well-known, changes with age. Compared to younger counterparts, healthy older adults have reduced random exploration, in which choices appear uninfluenced by the value of options.

To investigate the mechanism of reduced random exploration, reaction time and choices between two slot machines in the Horizon Task were modelled using the drift diffusion model (DDM). In the DDM there is an accumulation of evidence over time, until a boundary threshold for either the explore or exploit option is crossed. The DDM used here can distinguish between two different drivers of random exploration, changes in the signal-to-noise ratio (SNR), with which reward information is represented, and changes in the threshold required to make a decision.

We showed that reduced random exploration in older adults results from a higher decision threshold. Whilst older adults had a lower SNR than younger adults, which could lead to more mistakes, older adults actually performed slightly better in the Horizon task than younger adults. Together this suggests that the higher decision threshold could be a healthy aging adaptation, which is overcompensating for the less accurate choices that could result from a lower SNR alone.

## Introduction

The world's population is aging fast, such that by 2030 it is predicted that 1 in 6 people will be 60 years or older [1]. Older age comes with important decisions to make due to ever changing conditions, such as in health and finances [2]. Most research has focused on when decision making processes go awry in aging, such as risky gambling decisions in Parkinson's disease [3], yet, little is understood as to how decision making changes in healthy aging. Colloquially, the common lament that older adults are 'stuck in their ways' could indicate that healthy older adults engage differently with a particular type of decision, whether to explore or exploit.

To explore the unknown or exploit what we know well is a decision that must be balanced throughout our lives. Exploring yields information about which option is best but comes with the risk that the unknown outcome will be bad. Conversely exploiting is more likely to avoid a bad outcome but comes at the cost of less information. For instance, after years in academia, should you explore a new job opportunity in a field you know nothing about or exploit the job you are already in? Exploring a new career as a baker could lead to a happier life making beautiful cakes, like a highlight reel from the Great British Bake Off, but equally you could find baking a relentless grind of early mornings and low pay. In contrast, exploiting your current job in academia ensures a continuation of what you know, but you may forever wonder if you could have been happier baking cakes.

Classical aging theories suggest older adults choose to exploit more and explore less than younger adults, to adapt to current cognitive capabilities and emotional goals. Older adults have less capacity for cognitive control and so in the Cognitive Control Hypothesis older adults are hypothesized to favor exploitation over exploration because the cognitive costs of exploring are too high [4,5]. Furthermore, Socioemotional Selectivity Theory posits a shift to exploit behavior in older adults, in order to attain certain and immediate emotional and social goals, rather than exploring more uncertain, future-based goals. This is proposed to be due to a reduction in time before death,to profitably use information gained from exploring new options [6,7]. Therefore, both cognitive and emotional changes may be biasing older adults to exploitation over exploration, as seen in previous studies [8,9], although see [10].

While these theories of aging suggest that exploration may decrease with aging, more recent work suggests that different strategies for exploration change dissimilarly with age [11]. Previous work has shown that people use two distinct strategies to make explore-exploit decisions: directed exploration and random exploration [12]. Directed exploration is an explicit bias towards choosing more informative options. In contrast, random exploration is a 'noisy' choice selection, in which choices are less obviously tied to the value of options, appearing more random overall.

In both of these strategies, we assume that the decision is made by comparing the values of exploring and exploiting. In directed exploration, values are computed by adding an 'information bonus' to the expected short-term reward from the choice. This 'information bonus' approximates the long-term value of the information gained by exploration and, by increasing the value of the exploratory option, makes it more likely that we will explore. Conversely, in random exploration the decision to explore is driven by chance. In this strategy, 'noise' in the decision process, sometimes directs the individual away from choosing the option with the highest expected reward. Both exploration strategies appear to be under cognitive control because when it is more valuable to explore, such as when there is time to use the information or there is large uncertainty, both directed and random exploration increase [12].

Distinguishing between directed and random exploration is difficult in many explore-exploit tasks, but the Horizon Task has been specifically designed to do this (Fig 9). In this gambling task, participants must choose between two slot machines, which pay out a probabilistic reward based on varying Gaussian distributions. These slot machines vary in how much information participants have about them at the start in the form of 'instructed' trials, with more instructed plays leading to lower uncertainty about the machine's average payout and fewer instructed plays leading to more uncertainty. In addition, the overall balance between exploration and exploitation is changed by varying the horizon, i.e., how many choices participants have in that 'game,' with exploration favored when the horizon is long, and exploitation favored when the horizon is short. Manipulating information and horizon allows quantification of information seeking and behavioral variability, and consequently directed and random exploration [12].

Mizell *et al.* [11] used the Horizon Task to investigate levels of directed and random exploration in older adults. In this task, directed exploration is calculated as the difference in information seeking between the short and long horizon (Fig 9E). Mizell *et al.* found that older adults exhibited less information seeking overall but still increased their information seeking when the horizon was long. This suggests that horizon-dependent directed exploration is similar in younger and older adults. Random exploration is determined by the difference in behavioral variability between horizon conditions (Fig 9F). Older adults had similar baseline variability to younger adults, but did not increase this variability by as much when the horizon was long. This suggests that horizon-dependent random exploration is reduced in older adults.

In this paper we focus on the question of *how*, from a mechanistic perspective, random exploration is reduced in older adults. To do this we focus on a prominent theory of decision formation known as the drift diffusion model (DDM, Fig 1). In this model, a decision is made via the accumulation of "evidence" that is integrated over time [13,14]. This evidence denotes momentary information in favor of one decision or another such that the accumulated evidence increases if the evidence is in favor or one choice (e.g., exploration) and decreases if the evidence is in favor of the other choice (e.g., exploitation) [15–17]. To make this decision the accumulated evidence must cross an upper threshold to make one choice (e.g., explore) or a lower threshold to make the other choice (e.g., exploit) [17,18].

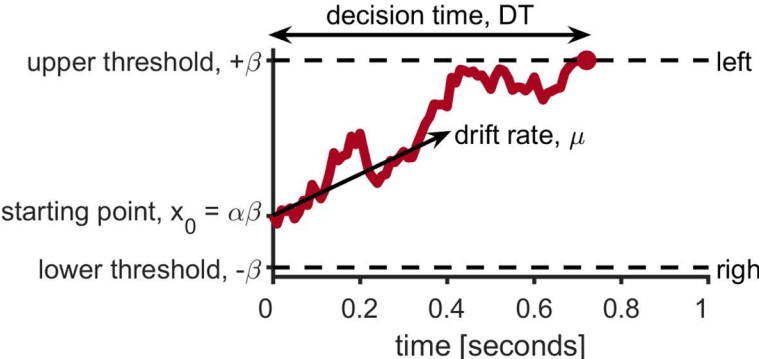

**Fig 1. Illustration of the drift diffusion model.** In this model a decision is made via accumulation of evidence over time until a threshold for a choice is met. The accumulation process starts from $X_0$, which represents the initial bias towards a particular option. Evidence is made up of a signal $\mu$, proportional to the difference in expected values between options, and noise with variance $c^2$. A decision is made when the accumulated evidence crosses a threshold $+/-\beta$. **Reproduced from Feng et al [17] licensed under CC BY 4.0.**

Critically, the evidence in the DDM is noisy – comprising a mixture of 'signal,' which tends to drift towards a preferred option (e.g., the high value option), and 'noise,' which reflects random fluctuations in the evidence [14,18]. The noise ensures that the resulting choices are somewhat random. Moreover, the influence of this noise on the decision process can be controlled in three different ways [18], suggesting three different mechanisms by which random exploration could be controlled and could change in aging.

First, and most obviously, choices could be made more random by increasing the amount of noise in the evidence. Likewise, decreasing the amount of signal in the evidence would also increase randomness by decreasing the ratio of signal to noise. Finally, and most subtly, choices could also be made more random by reducing the threshold required to make a decision. If the thresholds for exploring and exploiting are low, random noise in the evidence has a much greater chance of pushing the decision over the 'wrong' threshold.

Unfortunately, the three different mechanisms for controlling noise are indistinguishable at the level of choice. That is, identical changes in random exploration can be implemented by changing the noise, changing the signal, changing the threshold, or some combination of all three. Thus, previous work, such as Mizell *et al.*[11], which only focused on choice behavior cannot distinguish between the three mechanisms. However, as shown in Feng *et al.* [17] it is possible to distinguish between *two* of the mechanisms when we include response times in addition to choices. Specifically, we can distinguish between a change in threshold and a change in the *ratio* of the signal to noise (the signal-to-noise ratio, SNR). By modeling choices and response times, Feng *et al.* showed that random exploration in younger adults was primarily driven by a change in SNR and not a change in threshold [17]. Here we investigate whether reduced SNR, increased thresholds, or a combination of both, underlie the reduction of random exploration in older adults.

## Results

### Choice behavior in the Horizon Task

Before discussing response times, we briefly summarize the main findings from choices alone. These largely recapitulate the findings of Mizell *et al.* [11] with a larger sample size and a focus on the model-based (instead of model-free) analysis of behavior.

### Overall performance is similar for older and younger adults

A simple measure of performance on the Horizon Task is to look at the frequency with which participants choose the objectively correct bandit – that is the bandit with the highest generative mean, which we refer to as the "proportion

correct." By this measure both younger and older adults perform the task far above chance levels of 50% (younger: M = 75.1%, SD = 9.1%, t(140) = 32.813, p < 0.001; older: M = 76.3%, SD = 9.2%, t(156) = 35.688, p < 0.001) and there is no difference in the overall proportion of correct responding between the age groups (t(296) = -1.185, p = 0.237).

Breaking out the proportion correct by trial and horizon, there were no age differences in performance on horizon 1 (Fig 2A), but a significant age x trial interaction over the course of the horizon 6 games (Fig 2B, F(5, 1405) = 8.390, p < 0.001, $\eta^2$ = 0.03, 95% CI [0.01, 0.05]). Consistent with learning which option was best, both age groups increased their proportion correct over the course of the horizon 6 games. Younger adults had a significantly lower accuracy than older adults on trial 1 (t(578)=−3.941, p<0.001) and trial 2 (t(578) = -3.106, p=0.0020), but not later trials. The difference in accuracy between younger and older adults on the first free-choice trial in horizon 6 but not horizon 1 (age x horizon F(1, 296) = 24.609, p<0.001, $\eta^2$ = 0.08, 95% CI [0.03, 0.14]) seems to be driven by younger adults performing with lower accuracy in the first free-choice trial of horizon 6 compared to themselves on horizon 1 games (t(296)= 8.878, p<0.0001), in comparison to older adults who chose with approximately the same accuracy on the first trial in both horizon conditions.

### Older adults show less overall information seeking and less random exploration

Next, we analyzed behavior on the first free choice trial by computing the choice probabilities as a function of the difference in observed mean rewards for the two uncertainty conditions, two horizon conditions, and two age groups (Fig 3). In the unequal [1 3] information condition, older adults chose the more informative option less in both horizon conditions and for all reward values (Fig 3A and 3B). This is consistent with reduced information seeking in older adults. In the equal [2 2] information condition, older adults performed similarly to younger adults in horizon 1, but showed a steeper choice curve, consistent with less decision noise, in horizon 6.

The choice curves were quantified using a simple logistic model as previously described [17]:

$$p(left) = \frac{1}{1 + \exp\left(-\frac{\Delta R + A\Delta I + b}{\sqrt{2}\sigma}\right)}$$

(Eq 1)

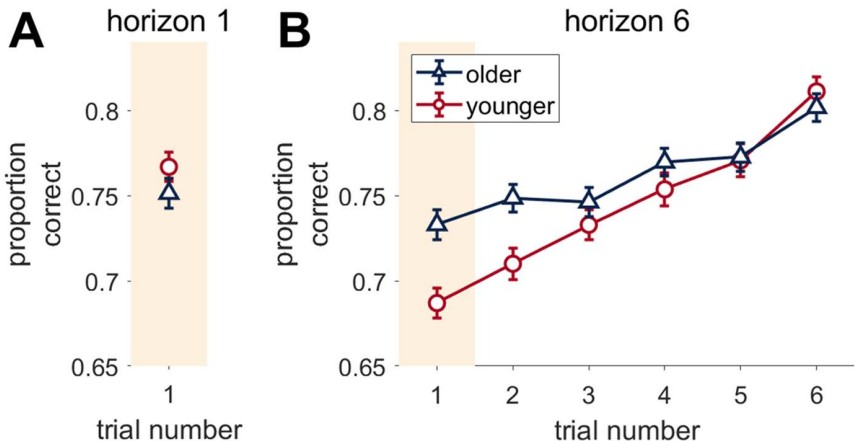

**Fig 2. Accuracy over trials in the Horizon task.** Younger and older adults have a similar proportion correct on horizon 1 trials, but older adults are significantly more accurate than younger adults on trial 1 and 2 of horizon 6 trials. Proportion Correct on horizon 1 **(A)** or horizon 6 **(B)** games. 'Proportion correct' is the proportion that the participants picked the bandit with the highest actual mean reward. Error bars are s.e.m.

PLOS Computational Biology

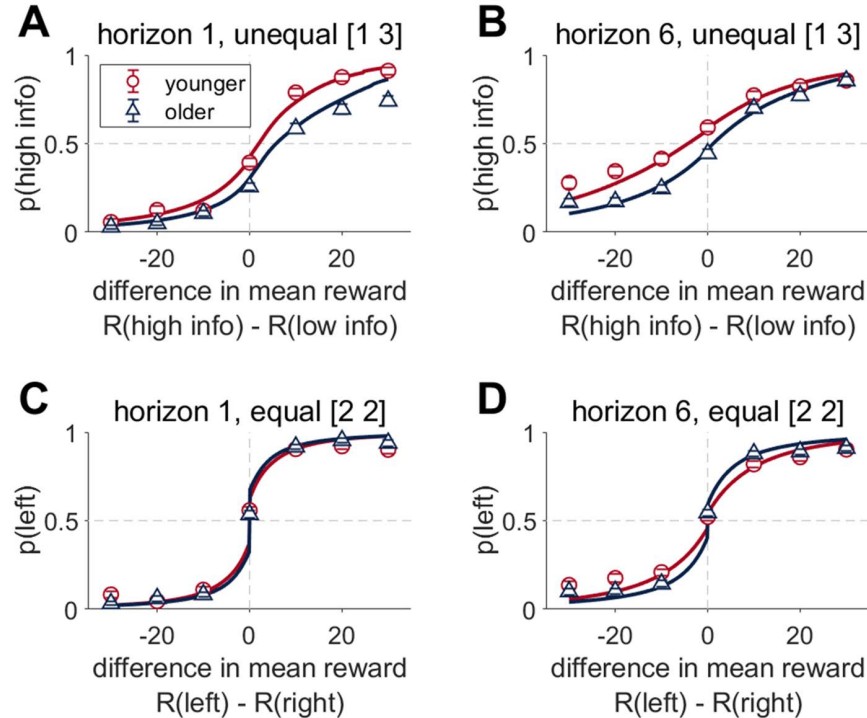

**Fig 3. Choice behavior in the Horizon Task on the first free choice trial.** Choice probability (p) as a function of the difference in observed mean reward **(A, B)** Under unequal conditions, older adults chose the more informative option less than younger adults under both Horizon conditions. The probability of choosing the more informative option (p(high info)) is shown as a function of the difference in observed reward between the more informative (R(high info)) and less informative (R(low info)) bandits. The more informative option is the option played only 1 rather than 3 times in the instructed unequal trials. (A) Horizon 1 games (B) Horizon 6 games. **(C, D)** Under equal conditions, older adults performed similarly to younger adults on horizon 1 but had a steeper curve on horizon 6, suggesting lower decision noise. For equal information conditions, the probability of choosing the left bandit is shown a function of the difference in observed mean reward between bandits on the left or right. (C) Horizon 1 games, (D) Horizon 6 games. Error bars are s.e.m.

where the observable task variables are $\Delta R$, the mean difference in reward $\Delta R = R_{left} - R_{right}$, $\Delta I$, the difference in information ($\Delta I = +1$ if the left option is more informative, $\Delta I = -1$ if the right option is more informative and $\Delta I = 0$ in the equal [2 2] condition). The free parameters are the information bonus ($A$), spatial bias (in favor of choosing the left bandit) ($b$), and decision noise ($\sigma$).

Optimal performance in the Horizon task has previously been calculated to have zero decision noise and an information bonus of 0 in horizon 1 and 3 in horizon 6. Previous reports show a consistent qualitatively similar horizon-mediated behavior in younger adults, although quantitatively, they often have a quantitatively higher information bonus and do show decision noise [12]. Here, both younger and older adults showed a larger information bonus in horizon 6 than horizon 1 (Horizon effect $F(1,296) = 87.0579$, $p < 0.0001$, $\eta^2 = 0.23$, 95% CI [0.15, 0.31], younger $t(296) = -6.817$, $p < 0.0001$, older $t(296) = -6.880$ $p < 0.0001$), consistent with directed exploration (Fig 4A). However, older participants had a lower information bonus than younger adults in both horizon 1 and 6 games (Age $F(1,296) = 37.9395$, $p < 0.0001$, $\eta^2 = 0.11$, 95% CI [0.05, 0.18], horizon 1 $t(521) = 6.678$, $p < 0.0001$, horizon 6 $t(521) = 7.537$, $p < 0.0001$), indicative of less overall information seeking in older compared to younger adults.

Consistent with random exploration, decision noise in horizon 6 was increased from horizon 1, in both younger and older adults, in both unequal (Age*Horizon $F(1,296) = 7.5879$, $p = 0.0062$, $\eta^2 = 0.02$, 95% CI [0.00, 0.07], younger $t(296) = -4.121$, $p < 0.0001$, older $t(296) = -4.655$ $p < 0.0001$) and equal (Age*Horizon $F(1,296) = 4.6084$, $p = 0.033$, $\eta^2 = 0.02$,

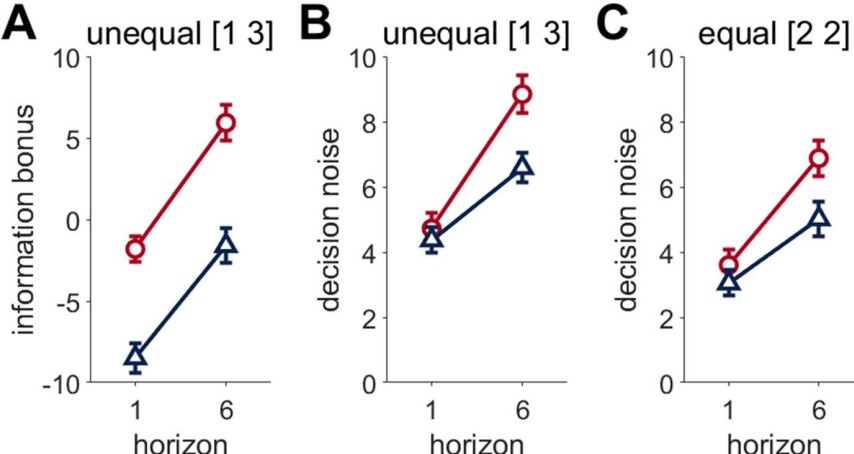

**Fig 4. Fit parameter values from the logistic model. (A)** Information bonus is increased with a longer horizon in both younger and older adults. However, older adults had a lower information bonus than younger adults under both horizon conditions, suggesting lower overall information seeking. **(B, C)** decision noise in the unequal (B) and equal (C) conditions was similar in horizon 1 in both age groups, but older adults had lower decision noise in horizon 6 than younger adults, suggesting older adults have less random exploration. Parameters are fit with the bias-noise-bonus model (eq 4). Error bars are s.e.m.

95% CI [0.00, 0.05], younger (t(296)=-7.413, p<0.001, older (t(296)=-4.702, p<0.0001) conditions. Yet, despite similar noise in horizon 1, younger adults have greater decision noise on horizon 6 than older adults in both conditions (unequal (t(488)=3.384, p=0.0043 and equal (t(430)=1.860, p=0.038, Fig 4B and 4C respectively) suggesting older adults have less random exploration than younger adults. Whilst younger adults are closer to the optimal information bonus in both horizons, the older adults had lower decision noise in horizon 6, which may lead to less errors and explain their overall higher proportion correct on earlier trials.

## Response times

### Older adults respond more slowly than younger adults, but with the same basic patterns

Overall, response times are significantly slower in older participants in both Horizon conditions, in both instructed and free choice trials (Age, horizon 1, F(1,296) =119.879, p<0.0001, $\eta^2$ = 0.29, 95% CI [0.21, 0.37](Fig 5A), horizon 6 F(1,296) =112.748, p<0.0001, $\eta^2$ = 0.28, 95% CI [0.20, 0.35] (Fig 5B)). However, both older and younger participants show the same qualitative pattern of response times in the Horizon task, in which participants are fast on the instructed trials (i1-4), and then significantly slower on the first free choice trial (i4 compared to trial 1: horizon 1, younger (t(1184)=-14.670, p<0.001, older t(1184)=-17.710, p<0.001) and horizon 6, (younger t(2664)=-19.210, p<0.001, older t(2664)=-30.465, p<0.001). This slowing is of a similar magnitude on trial 1 of both horizon 1 (Fig 5A) and horizon 6 (Fig 5B) games. In horizon 6 games, after the first free choice trial, the response times decrease over the remaining trials to end at a similar level to the instructed trials (no significant difference between i4 and trial 4 onwards in either younger or older adults).

Overall, this pattern suggests that both young and old participants wait to make the decision on the first free choice trial, rather than pre-empting. This suggests that it is valid to model the response times on trial 1 with a drift diffusion model.

### Younger and older adults show different modulation of response times by reward

Next, we looked at how response times vary according to the difference in reward on the first free choice trial (Fig 6A-6D). In all conditions, participants are slower for the harder decisions , i.e., when the difference in observed rewards is small. This reward-response time modulation is consistent with the decision being made on this first free choice trial.

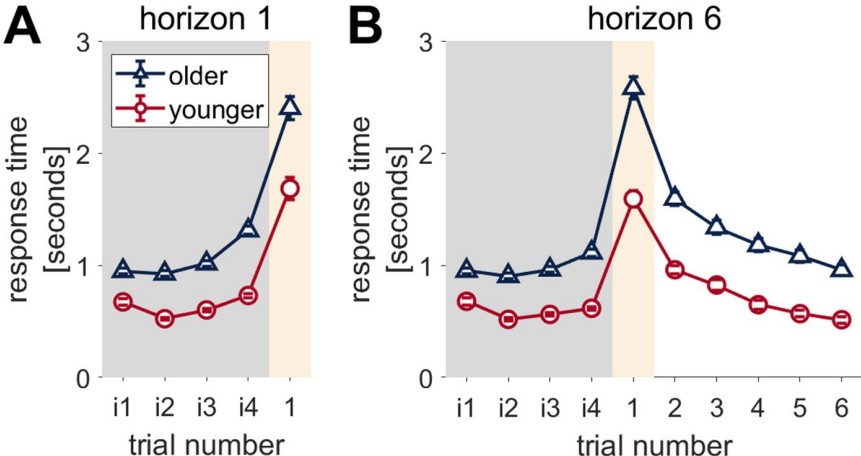

**Fig 5. Older adults have slower response times but show a similar qualitative pattern of response to younger adults.** Participants are faster on the instructed trials (i1- i4) than trial 1, suggesting that participants wait to make their decision on the free choice trial rather than deciding during instructed plays. **(A)** horizon 1 and **(B)** horizon 6 games. Instructed trials are denoted by 'i' before the number, number only denotes a free choice trial. *Error bars are s.e.m.*

However, the degree of this reward-response time modulation on the first trial appears to differ between younger and older adults (Fig 6A-6D). So to better quantify the effect of reward on response time, we fit a linear regression model to the reaction time. In this model, the assumption is that reaction time is given by:

$$RT = \beta_0 + \beta_R \alpha \Delta R + \beta_I \alpha \Delta I \qquad \text{(Eq 2)}$$

Where $\Delta R$ is the difference in mean reward and $\Delta I$ is the difference in information. $\alpha$ is the choice $+1$ for left and $-1$ for right and the regression coefficients $\beta_0, \beta_R$, and $\beta_I$ describe the baseline response time, linear effect of reward on response time and linear effect of information on response time respectively.

The baseline reaction time $\beta_0$ is slower overall in older compared to younger participants (Age, $F(1,296)= 70.525$, $P < 0.0001$, $\eta^2 = 0.19$, 95% CI [0.12, 0.27], Fig 6E). Younger participants show a decrease in baseline reaction time in horizon 6 compared to horizon 1 (Age*Horizon $F(1,296)= 25.673$, $\eta^2 = 0.08$, 95% CI [0.03, 0.14], younger H1vH6 ($t(296) = 6.974$, $p < 0.0001$)), whilst older participants' baseline reaction times do not differ between horizon 1 and 6 (Fig 6E).

Modulation of response time by reward, $\beta_R$, is significantly greater in horizon 6 compared to horizon 1 for both younger and older participants (Horizon $F(1,296) = 193.298$, $p < 0.0001$, $\eta^2 = 0.40$, CI[0.32, 0.47]. $\beta_R$ is similar for younger and older participants in horizon 1, but older participants have a significantly lower $\beta_R$ in horizon 6 than younger (Age*Horizon, $F(1,296) = 15.023$, $p < 0.001$, $\eta^2 = 0.05$, CI[0.01, 0.10], younger v older on horizon 6 $t(464) = -5.493$, $p < 0.0001$, Fig 6F), suggesting weaker modulation of response time by reward in older adults.

Modulation of response time by information, $\beta_I$, is lower in horizon 6 than horizon 1 (Horizon $F(1,296) = 69.7469$, $p < 0.0001$, $\eta^2 = 0.19$, CI [0.11, 0.27]). Whilst both younger and older participants show a decrease in modulation of response time by information in horizon 6 compared to horizon 1 (Younger $t(296) = 4.829$, $p < 0.0001$, Older $t(296) = 6.929$, $p < 0.001$), older participants' response times have a greater reliance on information than younger participants under both horizon conditions (horizon 1, $t(458) = 4.613$, $p < 0.0001$, horizon 6 $t(458) = 3.404$, $p = 0.0040$, Fig 6G).

Taken together, these results suggest that in addition to being slower overall, older adults' response times are modulated by reward and horizon in a different manner to younger adults. To better understand this pattern of behavior we now fit the drift diffusion model.

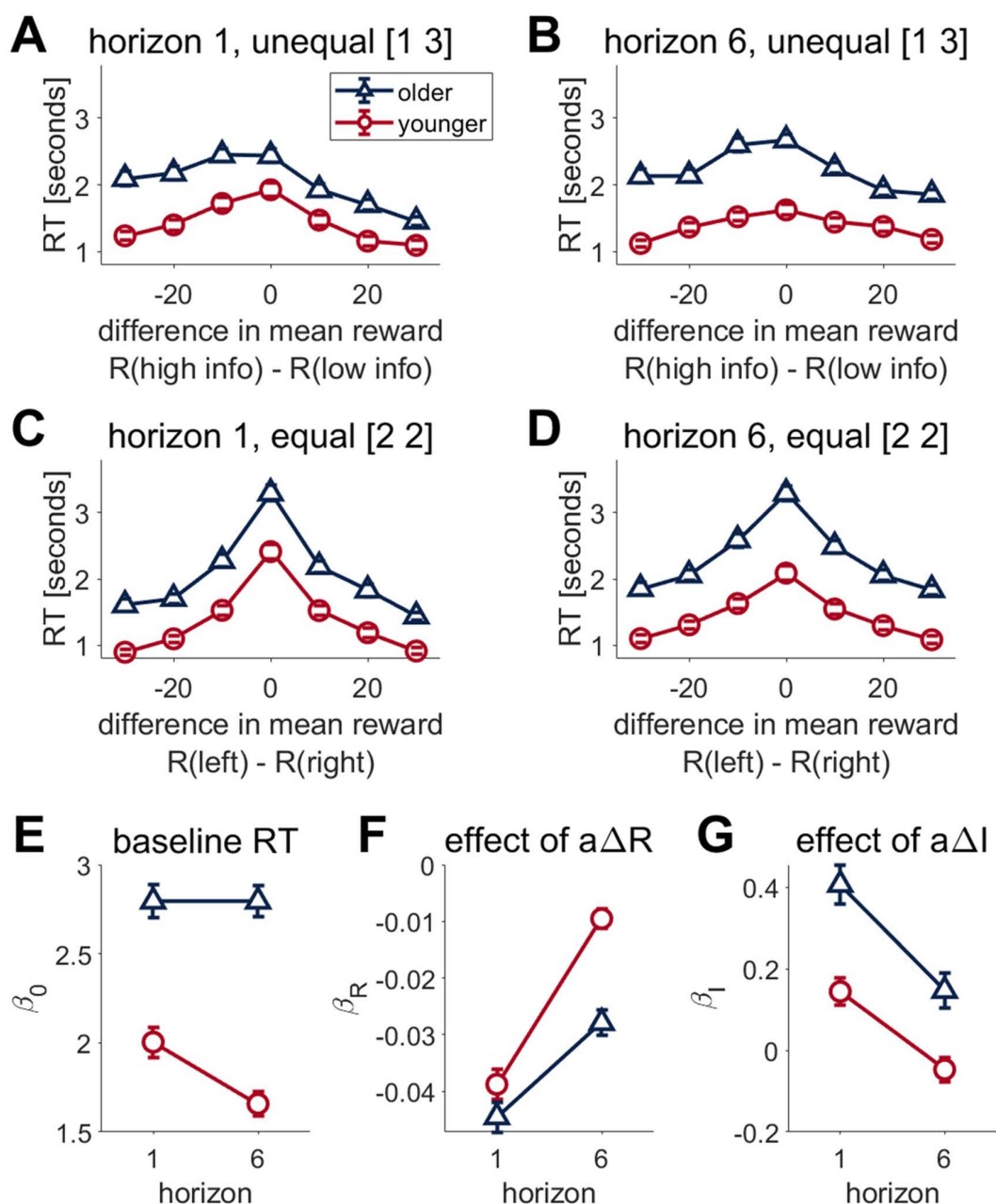

**Fig 6. Older adults show a different modulation of response time by reward. (A-D)** Response times on first free choice trial. (A,B) Response time as a function of difference in mean observed reward in unequal conditions in (A) horizon 1 and (B) horizon 6. Participants are slower for the harder decisions, i.e., when the difference in reward is small. Older adults are slower than younger adults but also have a different reward-response time relationship, especially when the difference between the R(high info) and R(low info) is negative. (C,D) In the equal condition, response time as a function of the difference between mean observed reward on the left and right bandits in (C) horizon 1 (D) horizon 6. Older adults are slower than younger adults but show a similar reward-response time relationship. **(E,F,G)** Linear regression analysis showing the (E) baseline response time, and the effects of $\alpha\Delta R$ (F) and $\alpha\Delta I$ (G) on response times. Baseline reaction time $\beta_0$ is higher in older adults and doesn't decrease for horizon 6, unlike younger adults. Older adults have a weaker modulation of response time by reward $(\beta_R)$ in horizon 6 and a higher modulation of response time by information $(\beta_I)$ in both horizon conditions. RT: response time, $\alpha$ is the choice, $\Delta R$ is the difference in mean reward and $\Delta I$ is the difference in information. Error bars are s.e.m.

## Drift diffusion modeling suggests that reduced random exploration in aging is driven by less flexibility in SNR and threshold

As shown in Fig 7, the drift diffusion model provides an excellent fit to the choice and response time data on the first free choice trial. The free parameters of this model are the 7 coefficients, $c_j^i$ ($i \in \{\mu, \beta, \alpha\}$; $j \in \{0, R, I\}$) and an additional non-decision time $T_0$. Each of these parameters are assumed to vary with Horizon, giving 16 free parameters.

Both the model and behavior show that younger adults use greater directed exploration in the unequal [1 3] condition in both horizon 1 and 6. Younger adults also show more random exploration in horizon 6 in the equal condition [2 2]. Compared to younger, older adults have consistently slower reaction times and modulation of reaction time by difference in reward is not as predominant.

Parameters from the drift diffusion model fit to behavior on the first free choice are shown in Fig 8 (and S1 Fig for distribution plot). This analysis reveals differences in several key parameters – threshold, non-decision time, and signal-to-noise ratios for reward and information – that appear to underlie the age difference in explore-exploit behavior.

Older adults have a higher baseline threshold than younger adults in both horizon 1 and 6 (Horizon*Age F(1,296) = 17.786, p<0.0001, $\eta^2$ =0.06, CI[0.02, 0.12], young v old on horizon 1 t(351) = 5.483, p<0.0001, horizon 6 t(351) = 7.959 p<0.0001, Fig 8A) with no reduction in threshold in horizon 6 unlike younger adults (t(296) = 5.957 p<0.0001) (Here in Fig 8A and shown previously [17]).

The non-decision time of older adults was also significantly greater compared to younger adults (Age F(1,296) = 97.9664, p<0.0001, $\eta^2$ =0.25, CI[0.17, 0.33], Fig 8B). Qualitatively, in younger adults the non-decision time slightly decreased in horizon 6 compared to horizon 1, whilst in older adults there was a small, non-significant increase (Fig 8B).

The effect of ΔR on SNR reward ($c_R^\mu \Delta R$) is reduced in horizon 6 compared to horizon 1 in both younger and older adults (Horizon, F(1,296) = 48.9180, p<0.0001, $\eta^2$ =0.15, CI[0.08, 0.22], young t(296)= 7.408, p<0.0001, old t(296) = 2.616, p= 0.046, Fig 8C). However, whilst both younger and older adults have a similar SNR reward in horizon 6, the older adults

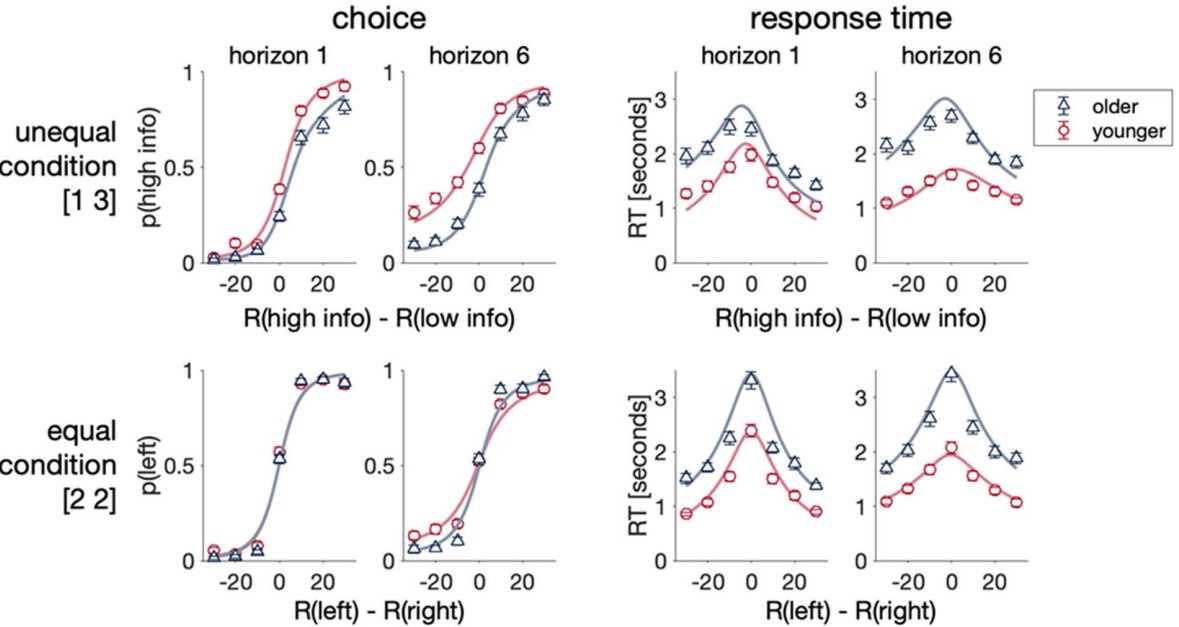

**Fig 7. The drift diffusion model (lines) recapitulates the main qualitative effects of both choice and response time data.** Posterior predictive checks comparing model fit (lines) with data (triangles, circles). Excellent agreement for both choice and response times for both younger and older adults. Error bars are s.e.m.

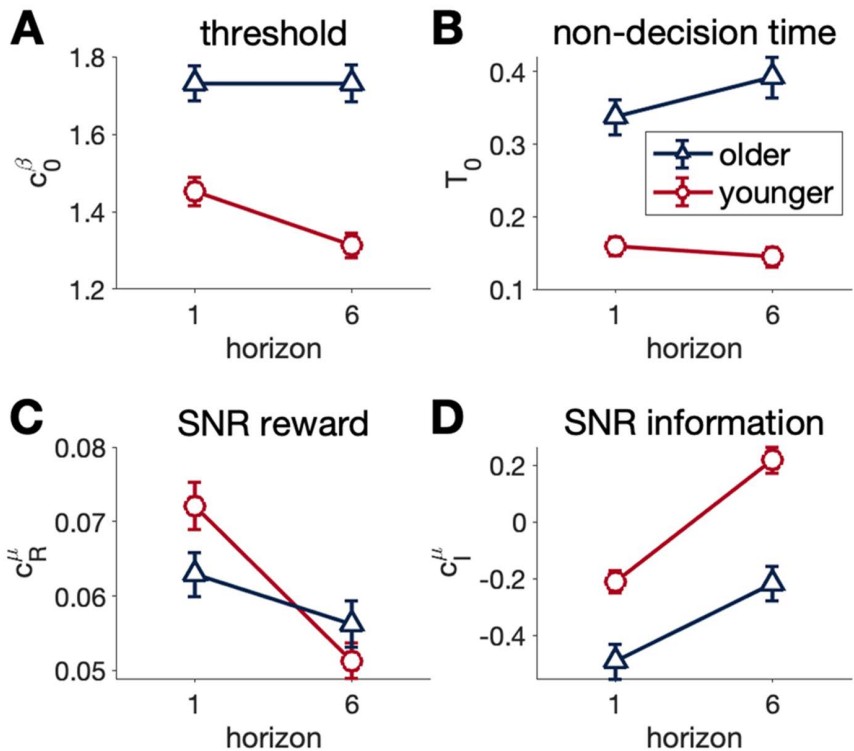

**Fig 8. Drift-diffusion model parameters in the Horizon task. (A)** Older adults have a higher threshold than younger adults **(B)** Older adults have a longer non-decision time than younger adults **(C)** The signal to noise ratio (SNR) for reward as a function of Horizon. Older adults have a lower reward SNR than younger adults in Horizon 1 but not 6. SNR decreases with Horizon for both groups, but more significantly with younger adults. **(D)** SNR for information, increases with Horizon in both younger and older adults. The SNR for information is lower in older adults under both Horizon conditions. Error bars are s.e.m.

have a significantly lower SNR in horizon 1 (Horizon*Age $F(1,296)$ = 12.8011, $p < 0.001$ $\eta^2$ = 0.04, CI[0.01, 0.09], younger v older $t(444)$ = -2.856, $p$ = 0.023, Fig 8C).

Younger and older adults both have an increased effect of $\Delta I$ on drift (SNR information) in horizon 6 compared to horizon 1(Horizon $F(1,296)$ = 103.1863, $p < 0.0001$, $\eta^2$ = 0.26, CI[0.18, 0.34], young $t(296)$ = -7.479 $p < 0.0001$, old $t(296)$ = -6.907 $p < 0.0001$, Fig 8D). Yet at both horizon 1 and 6, older adults have a lower SNR information than younger adults (Age $F(1,296)$ = 28.8558 $\eta^2$ = 0.09, CI[0.04, 0.16], Horizon 1 $t(484)$ = -4.261 $p$ = 0.0001, Horizon 6 $t(484)$ = -4.958 $p < 0.0001$).

The results of model fitting suggest that both SNR and threshold changes may underlie changes in behavioral variability associated with decreased random exploration in older adults. To show the behavioral variability is related to random exploration, we correlated noise in the DDM with noise indicative of random exploration in the logistic softmax model. We showed good correlation between approximation of noise in the DDM and logistic model ($r > 0.54$, $p < 0.0001$, S3 Fig, similar to before [17]).

To quantify how much an increase in threshold may compensate for higher SNR reward, we computed the softmax temperature with older adult average SNR reward and threshold and compared it to the temperature when the threshold is replaced with the average threshold for younger adults. There was an increase in temperature in both horizon 1 (18.5%) and 6 (29.4%), suggesting that older adults would have a decrease in accuracy were their threshold not higher than their younger counterparts (Table C in S1 Text).

We also assessed how DDM parameters change over the four blocks of the task. Younger adults significantly decrease their threshold over the task (Age*Block, (F(3,1419.2) = 2.22, p<0.001, $\eta^2$ =0.02, CI[0.00, 0.03], block 1 v block 4 t(1419)=4.58, p<0.0001), while older adults initially increase their threshold from block 1 –2 (t(1419) = -3.24, p= 0.0067, S5A and S5E Fig). Younger adults may also increase SNR reward in horizon 1 on block 3 and 4, while older adults do not (S5C Fig), although main effect of Age*Block did not reach significance.

To determine if a more parsimonious version of our model may be better, we conducted model analysis using leave one out cross validation (S6 Fig). Here we yoked SNR reward, threshold or both SNR reward and threshold across horizon conditions. We found that the full baseline model used here was the best fit for the majority of younger and older participants.

Taken together these results suggest that reduced random exploration in older compared to younger adults is primarily driven by two age-dependent factors 1) a higher threshold, leading to slower response times overall, and 2) a lesser decrease in both threshold and SNR reward for the longer horizon, leading to a smaller increase in decision noise.

## Discussion

Older adults exhibit reduced random exploration in the Horizon Task in comparison to younger adults [11], but choice behavior alone cannot elucidate what underlies this change. In the current work, we used a drift diffusion model (DDM) to fit response times and choices to better understand this age difference in random exploration. Fitting this model to the behavior of younger adults, we reproduce findings in [17] showing that random exploration in younger adults is primarily driven by horizon changes in the signal-to-noise ratio (SNR) in the encoding of reward rather than the threshold for making a decision. In older adults, horizon changes in both SNR and threshold are smaller, consistent with less horizon mediated change in random exploration. In addition, older adults have a longer non-decision time, higher threshold for decisions, and lower SNR in horizon 1. Taken together this suggests that random exploration is reduced in older age due to less adaptation of SNR and threshold with horizon alongside a higher response threshold overall.

In younger adults, results here and previously [17], show that the reduction in SNR for reward is largely responsible for the increase in random exploration in horizon 6 compared to horizon 1. Older adults also exhibit a decrease in SNR for reward in horizon 6 compared to horizon 1, but the magnitude of this decrease is smaller, with older adults having similar SNR in horizon 6, but lower SNR in horizon 1. Unfortunately, behavioral modeling with the DDM cannot distinguish whether the differences in the signal-to-noise ratio between older and younger adults are due to differences in signal or noise. Thus, the smaller decrease in SNR for reward from horizon 1 –6 in older adults could be due to the signal, in which older adults have a low signal in horizon 1 that cannot be decreased further in horizon 6, or the noise, in which the brains of older adults are inherently more 'noisy.' In keeping with this latter point, we saw a similar pattern of lower SNR for *information* (Fig 8D) in which older adults had a lower SNR for information in both horizon conditions. More generally, overall drift rate (of which SNR is a component) has been observed in older adults in a number of other studies including perceptual and memory tasks [14,19–22] and a lottery decision task [23], although not lexical decision making tasks [19], which may be related to the improvements in crystalized intelligence with age (see Table B in S1 Text). Encoding of reward magnitude has been reported to be diminished in older adults [23,24] which may reduce signal in SNR of reward. Overall, this suggests that the changes in random exploration with aging occur in the context of a more general decline in the encoding of task-relevant information in many, but not all tasks.

At the neural level, a prominent theory for decreased SNR in older adults is higher noise due to reduced neuronal dedifferentiation [25,26]. In older brains there is a decrease in both catecholaminergic modulation [27] and inhibitory GABAergic transmission [28,29]. As catecholamines, such as dopamine and noradrenaline, are involved in the fine tuning of neural responses to cognitive processes in an inverted U-shaped curve [30,31], a decrease from optimal levels in combination with reduced inhibition may lead to less specific neuron populations responding and hence 'neuronal noise'. At the neuronal population level, 'over-recruitment' of brain areas in older adults, or use of brain areas that are not also

used successfully for the same cognitive tasks in younger adults [32], has been suggested to be an extension of neuronal dedifferentiation. However, evidence for this is limited. Whilst reduced precision of brain areas activated is correlated with decreased performance in memory and perceptual tasks, age didn't appear to mediate this [25]. Further research is needed to determine if there would be greater noise on fMRIs in older adults than younger adults whilst doing the Horizon Task. This is feasible seeing as increased neuronal variability is seen in monkeys in an exploratory 'state' [33] as well as increased fMRI signal variability when younger adults randomly explore in a two-armed bandit task [34].

Threshold changes in the DDM can also underlie differences in random exploration. In younger adults, here and in [17], the baseline threshold decreases when it is more profitable to explore, here in horizon 6 compared to horizon 1. Threshold was significantly higher in older compared to younger adults, and additionally the threshold in older adults did not significantly decrease in horizon 6. In older adults, the higher threshold in the Horizon Task may be a compensation for the overall lower SNR, reducing errors that could result from less distinction between signal and noise. In other cognitive tasks, a higher threshold is also seen in combination with the faster drift rate in perceptual and memory tasks [23–27] and a lottery decision task [28] in older adults. Whilst this could be an example of a healthy adaptation to the faster drift rate with aging, the resulting reduction in random exploration may also be responsible for age-related cognitive deficits.

The reasons for this higher threshold could be both behavioral and physiological. For instance, physiologically, the higher threshold could be a result of functional connectivity changes. In favor of this, the higher threshold is also seen in lexical decision making tasks [19] where there is a slower drift rate, and so the threshold change would not be a potential dynamic compensation to prevent error. A suggested mechanism for higher thresholds in DDMs is decreased disinhibition of basal ganglia circuits, indirectly by the pre-supplementary motor area (pre-SMA) [35]. A causal role of the pre-SMA in modulating decision thresholds in healthy younger adults has been shown by inhibiting or activating the pre-SMA with TMS and seeing an increase or decrease respectively in thresholds in a random dot motion task [36]. Alongside diffusion tensor imaging findings of a decrease in white matter tracts between the pre-SMA and basal ganglia in older adults [37], it could be hypothesized that higher decision thresholds in aging are a result of decreased pre-SMA disinhibition of basal ganglia. Further TMS studies modulating pre-SMA prior to the Horizon Task are needed to investigate this question.

Behaviorally, the higher threshold could also result from a change in the speed accuracy tradeoff with age. Older adults have been found to prefer to go slower and be accurate, rather than fast with mistakes [38,39]. This corresponds well with our finding of a slightly better performance in the Horizon Task in older adults and other findings in which older adults perform better on more difficult memory tasks than younger adults [19]. This speed-accuracy hypothesis also concurs with findings that older adults are less risk taking for reward gains [40], as speed could be seen as a risk when the possible consequence is error. Although interestingly, the decrease in risk-taking behavior is not seen in older adults for tasks with reward loss [40]. It would be interesting to examine the effect on decision thresholds in a version of the Horizon task with possible reward loss, rather than purely maximizing gains. This could give us more information on the relative influence of behavioral versus physiological changes on decision thresholds in the Horizon Task.

The better performance of older adults compared to younger, despite a lower SNR, tentatively suggests that the higher response boundary could be an effective counterbalance to decreased SNR and so a healthy adaptation in aging. Although there was no significant correlation between SNR reward and threshold on an individual level (S4 Fig), suggesting this is a groupwise effect. Interestingly, older adults may have different adaptations to different decision making tasks. For instance, in the lottery decision making task the use of a starting time DDM (stDDM) was able to elucidate that older adults had a quicker start time for considering probability relative to reward magnitude in comparison to younger adults [23]. Here, in comparison, we did not see any change in initial starting biases between young and old. In a driving decision making task, older adults compensated for a lower drift rate with a lower boundary criterion than younger adults [41], an adaption not frequently seen [19]. Perhaps the motivation for a fast response for safe driving leads older adults to compensate for slower perceptual drift rates and non-response times, by lowering the threshold for response. In older adults, we don't see any change in thresholds between horizon conditions, in which motivations to explore or exploit are different.

Yet, we do see a slight reduction in threshold in younger adults in horizon 6 (here and in [17]), when exploration is preferable, suggesting these boundary thresholds are amenable to change with different motivations, but these motivations may change with age. The amount of planning that older adults do during free choice trials or otherwise before their response could also be an adaptation, which may underlie reduced random exploration in comparison to younger adults. We are unable to address this with our current model, but have developed a stochastic planning theory of explore-exploit decision making in which the samples are simulations of future choices and outcomes [42], and future work should also address this possible adaptation in explore-exploit decision-making in older age.

In conclusion, modelling response times with a drift diffusion model, suggested that a reduction in random exploration in aging is driven by higher response thresholds in older adults. The higher threshold may represent a healthy aging adaptation to prevent mistakes resulting from the reduced signal-to-noise ratio with which decision information is represented in the brain.

## Methods

### Participants

**Ethics statement.** All participants gave written informed consent before participating in the study. The study was approved by the University of Arizona Institutional Review Board (study title: "The explore-exploit dilemma in human decision making"; protocol number: 2107033856).

**Recruitment and screening procedure.** Data from part of this study sample (51 older and 32 younger adults) was previously reported in [11] where methods are reported in full. Briefly, all experiments were conducted at the University of Arizona in Tucson, USA. Participants were recruited locally through a variety of methods including adverts in the local newspaper, email mailing lists, at local events, and the psychology department website. Participants were screened either on the phone or using an online questionnaire, and primary exclusion criteria were neurological disorder or cognitive impairment. Some recruitment calls also screened for secondary exclusion criteria of actual or potential metal in body and medications or disorders which increase the likelihood of seizure. These secondary criteria were to ensure participants could progress to magnetic resonance imaging and transcranial magnetic stimulation sessions of the study, which are not reported here. 2 participants previously excluded in [11] for secondary criteria are included in this dataset because they met primary criteria. All participants were paid $20 per hour, which was not contingent on performance.

### Exclusion criteria

Participants were excluded from analysis if their Montreal Cognitive Assessment (MoCA) score was less than 26 (40 out of 201 older adults and 11 out of 153 younger adults), this left 157 older and 141 younger adults.

### Sample demographics

A Kolmogorov–Smirnov test revealed a statistically significant difference in the education levels between older and younger adults (D = 0.54, p < 0.001) with older adults spending approximately three years longer in education (older: M = 17.1, SD = 2.08; younger: M = 14.5, SD = 2.07). A Chi-Square test for Independence showed no significant difference in gender composition within or between each age group (p > 0.05, all comparisons) (Table 1).

Race for part of this sample was already reported in [11]. The race of the remaining sample is present in Table 2. There is a lower racial diversity in older compared to younger participants, as also seen in the other participants in our sample from [11].

### Experiment protocol

In session 1, participants completed a battery of neuropsychological tests taking between 2–3 hours. This consisted of the MoCA alongside tests of processing speed, episodic memory, working memory, crystallized intelligence, language fluency,

**Table 1. Sample demographics for old and young participant groups.**

| Characteristic | Older (N = 157) | Younger (N = 141) |
|---|---|---|
| **Age** | 69.2 (65-74) | 21.4 (18-30) |
| **Gender** | | |
| F | 95(61%) | 85(60%) |
| M | 62(39%) | 55(39%) |
| Non-binary or other | 0(0%) | 1(0.7%) |
| **Education (in years and qualification equivalent)** | | |
| Up to 12 (High School) | 7(4.5%) | 33(23.4%) |
| 12-16 (Bachelors) | 60(38.2%) | 85(60.3%) |
| 16-18 (Masters) | 66(42.0%) | 18(12.8%) |
| 18-20 (PhD/JD/MD) | 20(12.7%) | 4(2.8%) |
| >20 years | 2(1.3%) | 0(0%) |
| Didn't answer | 2(1.3%) | 1(0.7%) |

**Table 2. Race of participants not reported in Mizell *et al.* [11].**

| Race | Older | Younger |
|---|---|---|
| White | 97 (95.1%) | 73 (70.2%) |
| Black/African American | 1 (0.98%) | 5 (4.8%) |
| Asian | 1 (0.98%) | 19 (18.3%) |
| Native American/ Alaska Native | 0 (0%) | 2 (1.92%) |
| Multiracial | 0 (0%) | 3 (2.9%) |
| Other | 1 (0.98%) | 1 (0.96%) |
| Prefer not to say | 2 (1.96%) | 1 (0.96%) |

and executive function (see Table A for individual tests used and Table B for results, in S1 Text). Participants previously reported in [11] then did the Horizon Task, whilst the rest of the participants were invited back for a second session. This second session started with the Horizon Task and then participants completed other decision making tests not reported here.

## Horizon task

The Horizon Task [12] was used to measure explore-exploit behavior (Fig 9). Participants completed 144 'games' (16 practice games and 128 test games), in which they chose between two slot machines/one-armed bandits over a varied time horizon. Completing these games took approximately 30–45 minutes.

In an individual game, the participant must initially complete 4 'instructed trials' (Fig 9A) which set up either unequal [1 3] - they see 1 draw from one bandit and 3 draws from the other bandit, or equal [2 2] information conditions- they see 2 draws from each bandit (Fig 9D left and right respectively). The participant then gets a 'free-choice', between the two bandits either in the Horizon 1 condition- in which there is 1 free-choice (Fig 9B), or Horizon 6, in which there are 6 free-choices (Fig 9C). When chosen, each bandit 'pays out' a reward in points, sampled from a Gaussian distribution, with a standard deviation of 8 points centered on the mean. In each game, the two bandits have different means of the Gaussian, and these are initially unknown to the participant, although they are informed in the instructions that one bandit always has a higher payoff than the other. To maximize points, the participant needs to balance exploring which bandit is best, with exploiting the bandit with the higher average payoff. Mean reward for each bandit varies from game to game. The

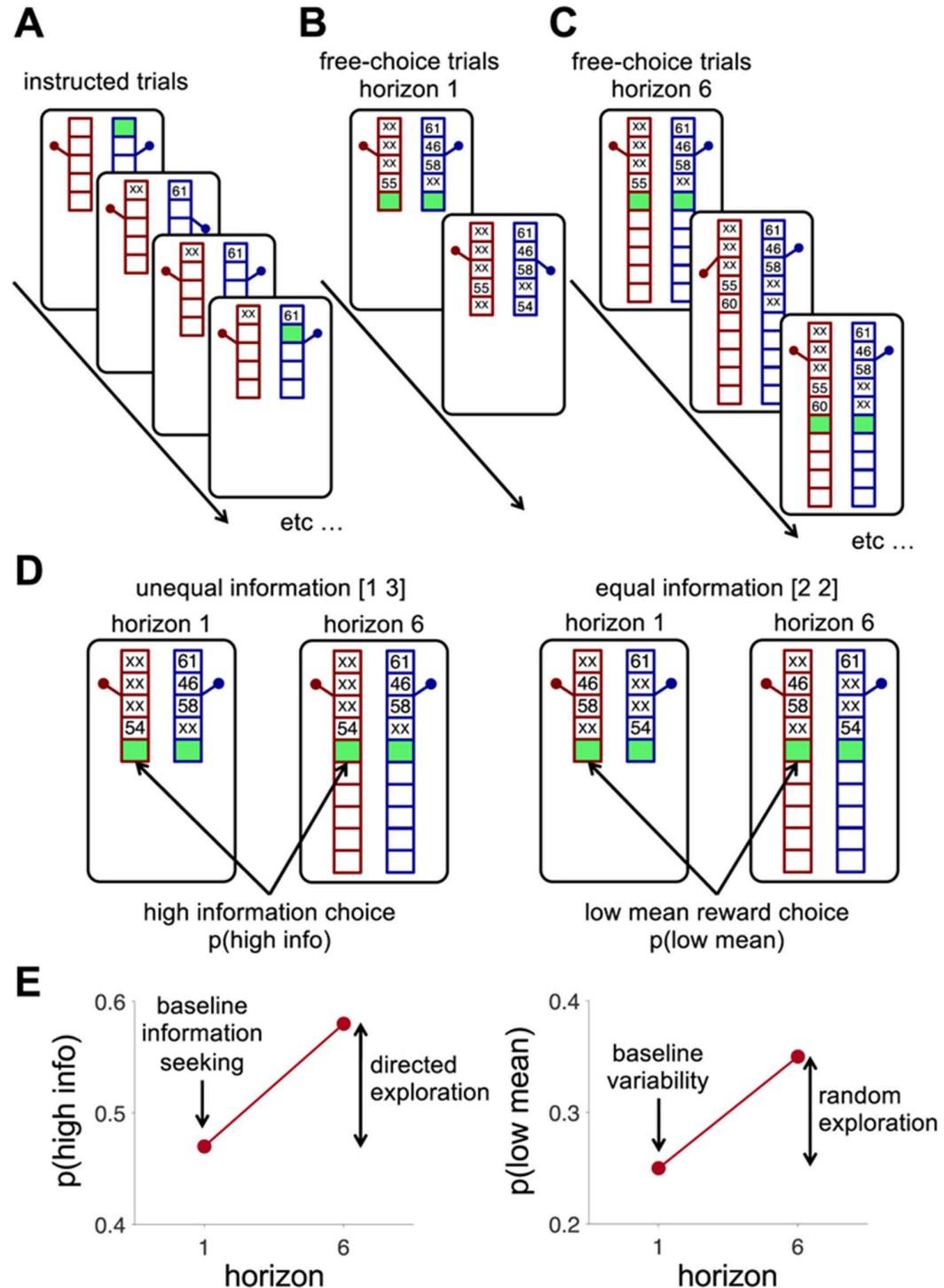

**Fig 9. The Horizon Task. (A)** Timeline of the instructed trials for a horizon 1 game. In these trials, a green box appears inside the bandit participants are required to choose. After a bandit is chosen, the reward is displayed, and the next trial begins. Instructed trials for horizon 6 games are similar, with the only difference being the length of the bandits which have slots for 10 trials. Participants play four instructed trials in both the horizon conditions. **(B, C)** After the instructed trials, participants have free choices between the two bandits for the rest of the game, which lasts either 1 trial (horizon 1 games, B) or 6 trials (horizon 6 games, C). Thus, one game lasts 5 trials (4 instructed, 1 free) in the horizon 1 condition and 10 trials (4 instructed, 6 free) in the horizon 6 condition. Overall, participants played 144 games. **(D)** The key analysis in the Horizon Task focusses on the first free-choice trial. On this trial,

the instructed trials set up one of two information conditions: an unequal, or [1 3], condition in which one bandit has been played once and the other three times (left), and an equal, or [2 2], condition in which both bandits have been played twice (right). **(E)** Choosing the high information option in the unequal condition quantifies baseline information seeking in horizon 1, while the change in p in horizon 6 quantifies directed exploration. Choosing the low mean reward options in the equal condition quantifies baseline behavioral variability in horizon 1, while the change in horizon 6 quantifies random exploration. This figure is reproduced from [11].

mean of one bandit was always 40 or 60 points (randomly counterbalanced across games) whilst the second bandit mean was offset relative to the mean of the other bandit by plus or minus 4, 8, 12 or 20 points.

There are two key manipulation conditions in the Horizon Task. The first is the amount of information (number of draws) from each bandit in the instructed trials. The unequal information condition enables assessment of information seeking as the probability of choosing the more informative option – the bandit played only once during the instructed plays (p("high info"), Fig 9E, left). The equal information condition allows assessment of behavioral variability as the frequency of mistakes –choosing the option with lower observed mean reward during the instructed trials (p("low mean"), Fig 9E, right).

The second manipulation is the eponymous 'horizon', or number of trials in each game. The horizon determines the value of exploration. A short horizon (1 trial) means that exploration has no value, because there is no opportunity to use new information in the future. However, in the long horizon condition (6 trials), exploring may gain valuable information, which can be used to guide decisions between bandits in the subsequent trials. The horizon manipulation enables distinction between baseline information seeking and directed exploration, as well as behavioral variability from random exploration. In horizon 1, we define baseline information seeking as p("high info") and baseline behavioral variability as p("low mean"). Conversely, we define directed exploration as the change in p("high info") between horizon 1 and horizon 6, and random exploration as the change in p("low mean") (Fig 9D & 9E).

### Drift diffusion model

Response times were modelled using an adaptation of the drift diffusion model (DDM). Introduced by Ratcliff [13], the original DDM has been used to model a variety of two-alternative forced choice paradigms [43–45]. More recently, the DDM has been used to study value-based decisions both in the Horizon Task [17] and other similar tasks , e.g., [15,16,46]. The model used here is based on that used in Feng *et al.* [17] and is an adaptation of what is commonly called the 'simple' or 'pure' DDM [15,18].

At every instant, the model encodes a relative value signal ($X$) representing the accumulated 'evidence' favoring the hypothesis that the left bandit has a higher value ($X > 0$) than the bandit on the right ($X < 0$). This relative value signal evolves according to a simple stochastic differential equation, written in Itô form as:

$$dX(t) = \mu dt + cdW(t) \tag{Eq 3}$$

where $\mu dt$ is a drift rate representing the average change in evidence supporting a left ($\mu > 0$) or right ($\mu < 0$) response and $cdW(t)$ is Gaussian distributed 'white noise' with mean 0 and variance $c^2 dt$.

A choice is made when the relative value crosses a threshold at +β for left and -β for right. Non-decision time ($T_0$) was fixed during an initial response time period, in which no accumulation is happening (i.e., $X(t)$ does not change for $t \in [0, T^0]$).

Finally, the accumulation starts at some initial state of evidence: $X(0) = X_0$ which we usually write as $X_0 = \alpha\beta$ where $-1 \leq \alpha \leq 1$. In this context, $\alpha$ is the `bias'. It is commonly known that one of $\mu$, $\beta$, $c$ can be fixed without changing the model's response time distributions [18], we thus fix $c = 1$. Our formulation of the simple/pure DDM then has 4 parameters: $\mu$, $\beta$, $\alpha$, $T_0$. Our modeling effort, then, is to incorporate the elements of the Horizon Task into these key parameters.

To model behavior on the first free-choice of each game, we assume that the drift rate, threshold, and bias, can all vary with difference in reward $\Delta R$ and difference in information $\Delta I$. Thus, we write:

$$\mu = c_0^\mu + c_R^\mu \Delta R + c_I^\mu \Delta I$$

$$\alpha = 2L\left(c_0^\alpha + c_R^\alpha \Delta R + c_I^\alpha \Delta I\right) - 1$$

$$\beta = c_0^\beta \tag{Eq 4}$$

where $L$ is a logistic link function (main text, Equation 2). This yields 7 free parameters to describe the baseline value, effect of reward and effect of information on each of drift and bias. Combined with the non-decision time, $T_0$, this gives us 8 free parameters that we fit to each horizon condition, giving 16 free parameters overall.

### Fitting the drift diffusion model

The drift-diffusion model was fit to participant choices and response times using a maximum likelihood approach. This approach centered on the method of [47] for fast and accurate computation of the first passage time distribution of the drift-diffusion process. Fits were performed in Matlab, using the fmincon function.

Parameter recovery was also performed by fitting simulated data. 298 participants worth of data were simulated using the maximum likelihood approach and the same parameters as the real data. Recovered parameters were then compared to the ground truth parameters from simulation (S2 Fig).

All code and data used to reproduce the figures and analysis are available in the OSF repository: https://osf.io/upn8t

### Statistics

For all analyses, a linear mixed-effect model was used to examine the effects of age and horizon (in addition to trial, where appropriate) (equation 3)

$$Y_{ij} = \beta_0 + \beta_1(Age)_i + \beta_2(Horizon)_j + \beta_3\left(Age_i \times Horizon_j\right) + u_i + \in_{ij} \tag{Eq 5}$$

In this model, $Y_{ij}$ represents the dependent variable for participant $i$ at horizon $j$, with $\beta_0$ as the fixed intercept, $\beta_1$ and $\beta_2$ the main effects of Age and Horizon, and $\beta_3$ the interaction. A random intercept, $u_i$, for each participant to account for repeated measures across conditions and $\in_{ij}$ represents the residual error, accounting for variability not explained by the fixed effects and random intercept.

The significance of each factor and interaction was assessed using F-tests, partial eta-squared ($\eta^2$) and 95% confidence intervals (CI). Where significant effects were found, post-hoc pairwise comparisons of estimated marginal means were conducted, with a Tukey test for multiple comparisons. All statistical analyses were carried out in R (version 4.4.1).

### Supporting information

**S1 Text. Table A.** Cognitive abilities and tests used for their assessment. If the test was part of a composite assessment, then the composite assessment is in italics. Abbreviations: WAIS IV: Wechsler Adult Intelligence Scale IV, RAVLT: Rey Auditory Verbal Learning test, NAART: North American Adult Reading Test, MINT: Multilingual Naming Task. **Table B.** Comparison of neuropsychological scores for old and young participants (excluding those who failed the MoCA). Statistics presented are means with lower and upper quartiles in parentheses. P-values correspond to Wilcoxon rank-sum test between the two groups. **Table C.** Pearson correlation coefficients (r) and significance values (p) between simulated and fit DDM parameters. **Table D.** Softmax temperatures. Older adult has the average SNR reward and threshold for older adults in this study. Older adult with younger adult threshold has average older adult SNR reward and average younger

adult threshold. The increase in temperature in both horizon conditions suggests an increase in noise and so a likely decrease in accuracy.
(DOCX)

**S1 Fig. Drift-diffusion model parameters in the Horizon task, same data as fig 8 but violin plot for distribution.** (A) Older adults have a higher threshold than younger adults (B) Older adults have a longer non-decision time than younger adults (C) The signal to noise ratio (SNR) for reward as a function of Horizon. Older adults have a lower reward SNR than younger adults in Horizon 1 but not 6. SNR decreases with Horizon for both groups, but more significantly with younger adults. (D) SNR for information, increases with Horizon in both younger and older adults. The SNR for information is lower in older adults under both Horizon conditions. * $p < 0.05$, ****$p < 0.0001$ between age group differences, × $p < 0.05$, ×××× $p < 0.0001$ within age group, between horizon differences.
(TIF)

**S2 Fig. Parameter recovery for the full model with maximum likelihood fits for horizon 1 (blue) and horizon 6 (red) games.**
(TIF)

**S3 Fig. Comparison between noise estimated from the logistic model and approximate noise from the DDM.** 8 subjects, who had negative drift rate parameters, were excluded from this analysis, leaving 290 participants.
(TIF)

**S4 Fig. Correlation analysis between SNR reward and threshold.** There was no significant correlation between SNR reward and threshold in either younger or older participants in either horizon 1 or 6. SNR: signal to noise ratio.
(TIF)

**S5 Fig. Younger adults seem to increase their SNR reward in Horizon 1 during block 3 and 4, while older adults do not.** Younger adults also decreased their threshold from block 1 –4, while older adults again did not. This suggests that significant differences in SNR reward and threshold may be driven by learning effects. Horizon 1: A) Threshold B) Non-Decision Time C) SNR reward, D) SNR information. Horizon 6: E) Threshold, F) Non-Decision Time, G) SNR reward, H) SNR information. All error bars are s.e.m.
(TIF)

**S6 Fig. To determine whether as much flexibility as we have in the current model (baseline) is needed, we conducted model comparison with threshold yoked (yoke $c_0^{\beta}$), SNR reward (yoke $c_R^{\mu}$) or both threshold and SNR reward (yoke $c_0^{\beta}+c_R^{\mu}$) yoked across Horizons.** Model comparison was performed using leave-one-out cross validation. The y-axis shows the percentage of participants best fit by each model. The baseline model was the best model for the highest percentage of both younger (A) and older (B) participants.
(TIF)

## Author contributions

**Conceptualization:** Robert C Wilson.

**Data curation:** Caroline Emily Phelps, Meghna Sreeram, Victoria D Antoniou, Larissa E. Oliveira, Jack-Morgan Mizell, Robert C Wilson.

**Formal analysis:** Caroline Emily Phelps, Robert C Wilson.

**Funding acquisition:** Ying-hui Chou, Gene E Alexander, Robert C Wilson.

**Investigation:** Caroline Emily Phelps, Alec E Frisvold, Meghna Sreeram, Victoria D Antoniou, Larissa E. Oliveira, Sierra R. Tooke, Yan Z. Lu, Anna C. Lemon, Joseph U Fraire, Alexe G. Delval, Betsy K. Smith, Devynn H. Spangenberg, Madelien N. Mithelavage, Michelle N. Ngo, Elizabeth M. O. Keller, Liliana J. Isaac, Sara A. Harader, Jack-Morgan Mizell, Siyu Wang, Waitsang Keung, Mary-Kathryn Franchetti.

**Methodology:** Waitsang Keung, Mark H Sundman, Ying-hui Chou, Gene E Alexander, Robert C Wilson.

**Project administration:** Caroline Emily Phelps, Alec E Frisvold, Robert C Wilson.

**Resources:** Robert C Wilson.

**Supervision:** Caroline Emily Phelps, Jack-Morgan Mizell, Robert C Wilson.

**Visualization:** Caroline Emily Phelps, Robert C Wilson.

**Writing – original draft:** Caroline Emily Phelps.

**Writing – review & editing:** Robert C Wilson.

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
