## [Decision Letter · Decision Letter 0]

22 Apr 2025

PCOMPBIOL-D-25-00273

The dynamics of explore-exploit decisions suggest a threshold mechanism for reduced random exploration in older adults

PLOS Computational Biology

Dear Dr. Phelps,

Thank you for submitting your manuscript to PLOS Computational Biology. After careful consideration, we feel that it has merit but does not fully meet PLOS Computational Biology's publication criteria as it currently stands. Therefore, we invite you to submit a revised version of the manuscript that addresses the points raised during the review process.

You will find the comments and questions of 3 experts in the field who have evaluated your manuscript and provided constructive criticism. In particular, reviewers #1 and #2 raised question about model comparison with degenerated version of your model to explore more parsimonious variants of the DDM as well as an integration with reinforcement learning models. Addressing these questions is crucial and will raise the quality of the modeling in your manuscript. In addition, reviewer #2 raised an issue about "conceptual ambiguity" in how the DDM maps onto random exploration. Resolving these theoretical questions and explaining how the DDM maps explore-exploit decision, which imply different choice alternatives between trials will also improve the clarity of the manuscript. Finally, exploring the relationship between DDM parameters and neurocognitive variables (as raised by reviewer #3) will help to contextualize the results.

Please submit your revised manuscript within 60 days Jun 20 2025 11:59PM. If you will need more time than this to complete your revisions, please reply to this message or contact the journal office at ploscompbiol@plos.org. Please include the following items when submitting your revised manuscript:

We look forward to receiving your revised manuscript.

Kind regards,

Jan Gläscher, PhD

Academic Editor

PLOS Computational Biology

Hugues Berry

Section Editor

PLOS Computational Biology

**Journal Requirements:**

At this stage, the following Authors/Authors require contributions: Caroline Emily Phelps, Alec E Frisvold, Meghna Sreeram, Victoria D Antoniou, Larissa Elia Oliveira, Sierra Rose Tooke, Yan Zhi Lu, Anna Catherine Lemon, Joseph U Fraire, Alexe G, R Delval, Betsy Kate Smith, Devynn Hailey Spangenberg, Madelien Nicola Mithelavage, Michelle Nhi Ngo, Elizabeth Margaret O'Connor Keller, Liliana Jade Isaac, Sara Arghiani Harader, Jack-Morgan Mizell, Siyu Wang, Waitsang Keung, Mark H Sundman, Mary-Kathryn Franchetti, Ying-hui Choi, Gene E Alexander, and Robert C Wilson. Please ensure that the full contributions of each author are acknowledged in the "Add/Edit/Remove Authors" section of our submission form.

Potential Copyright Issues:

i) Regarding Figure 9 , thank you for stating that it "is reproduced from [11]."  According to the APA website, the authors still need permission to reuse figures from the journal.  Please provide written permission to reuse the figure.

6) Please ensure that the funders and grant numbers match between the Financial Disclosure field and the Funding Information tab in your submission form. Note that the funders must be provided in the same order in both places as well. Currently, " R01 AG061888 " is missing from the Funding Information tab.

**Reviewers' comments:**

Reviewer's Responses to Questions

Reviewer #1: In this paper, Phelps and colleagues use a large data set of younger and older adults in combination with drift-diffusion modeling to generate new insights into aging effects on explore-exploit tradeoffs. Key to the innovation of this study is the use of response times in the Horizons task (a well-established assay of exploratory behavior). Their model-based results show that older adults exhibit reduced random exploration through a combination of lower drift rate and higher threshold.

The paper is well-written and the methods are rigorous. I appreciated the use of an unusually large data set for this kind of study, along with parameter recovery and predictive checks to establish the plausibility of the empirical results. The agreement beteween the data and model shown in Fig. 7 is impressive.

Major comments:

- 8 free parameters fit to each condition for each subject separately seems like a lot, given that there are only at most a few hundred trials per condition (and much fewer in the horizon=1 condition). The parameter recovery results in the supplement are convincing, so I'm not worried that parameter values can be fit accurately. I'm mainly wondering whether have so much flexibility is necessary to fit the data. This is a model comparison question (with a corresponding model recoverability question as well). For example, if you constrain either drift rate or threshold to be the same across conditions, can you still capture the data? Is this more parsimonious model preferred by model comparison?

- In the Introduction, the study is described as 'mechanistic', but I think it's somewhat implausible that the DDM, at least in the way it's used here, is the mechanistic basis for behavior in this task. To apply the DDM, the authors assume that there are relative reward and relative information signals driving the accumulator. This is similar to standard assumptions in applications of DDMs to value-based decision tasks. I'm ready to believe that some sort of evidence accumulation determines choices and RTs in such tasks. What this misses in the Horizon tasks specifically is the fact that one aspect of RTs is (hypothetically) a planning process, which is not modeled here. It could be the case that older adults do less planning, or their planning is more stochastic. This would be closer, in my view, to a truly mechanistic account (such a planning account is implied by the use of dynamic programming to find the optimal solutions in the original Wilson et al. paper, but this point seems to have been lost in the subsequent literature). I don't expect the authors to take on this kind of modeling project, but it's perhaps worth mentioning.

Minor comments:

- The paper sometimes refers to 'signal-to-noise ratio' (SNR) and sometimes to 'drift rate'. My understanding that these are the same. I suggest that, to avoid confusion, the authors adopt only one of these terms (I would say 'drift rate' is more standard).

- Please make sure to specify the error bars in all figure captions.

- This is a small point which the authors are welcome to take or leave: currently the paper is not as concise as it potentially could be because all the empirical results are shown once, and then shown again along with the model fits. If the authors wanted, they could compress this to only show them once together.

Reviewer #2: The manuscript examined whether exploration-exploitation patterns change as a function of age using the drift-diffusion model framework. Younger adults (ages 20-30) and older adults (ages 60-70) completed the Horizon Task. In this task, participants choose between two bandits one or six times. Before free choice trials, participants are presented with either balanced information (two samples from each option) or unbalanced information (one sample from one option, three from the other). The results indicated that older participants demonstrated less random exploration than younger ones and exhibited slower response times. Analysis of these findings using the drift-diffusion model revealed that older adults had a lower signal-to-noise ratio (SNR) but compensated with a higher decision threshold, resulting in higher accuracy. The researchers interpreted this pattern as an adaptive compensatory mechanism rather than simply reflecting cognitive decline.

The manuscript has several strengths: it addresses an interesting and important research question and the experimental paradigm has empirical foundations for investigating explore-exploit decisions. Additionally, the analysis of both choice behavior and response times enables a deeper examination of the cognitive mechanisms underlying exploration. The use of the drift-diffusion model framework allows to distinguish between competing explanations (reduced SNR vs. increased threshold) for age-related changes in exploration. Furthermore, the findings connect research on exploration-exploitation with cognitive aging. However, in my opinion, the manuscript suffers from conceptual ambiguity and several critical analyses are missing.

Main comments:

1. Conceptual ambiguity – The authors state in the abstract that "Reduced random exploration in aging could therefore be caused either by an increase in signal-to-noise ratio or an increase in decision threshold in older adults". However, I feel that there is a profound conceptual ambiguity in this statement. Within the standard drift-diffusion model framework, participants accumulate evidence based on drift rate, with the "correct" choice determined by the drift direction. In this theoretical framework, there is no conceptual space for "random exploration" – deviations from the drift-favored option are (by definition) errors resulting from noise. However, in the Horizon task a seemingly "suboptimal" choice provides valuable information that may help in subsequent decisions (same as in reinforcement learning frameworks). What appears as "noise" in a single-trial perspective may actually represent information-gathering behavior in a multi-trial context.

The authors attempt to address this by modeling only the first free choice, but this approach does not fully resolve the underlying theoretical tension: when restricted to a single decision point, the DDM inherently treats exploratory deviations as noise-induced errors. Indeed, by employing the DDM in isolation, the authors simplify the complexity of the task and can investigate choice and response time simultaneously. However, an important question that arises is whether the DDM in isolation (applied only to the first choice) is the appropriate theoretical framework for modeling exploration in this task? A possible more theoretically coherent approach would integrate reinforcement learning with the DDM framework.

2. Alternative models – The research findings rely exclusively on a single modeling approach. Comparison across multiple model variants would significantly strengthen the robustness of the conclusions. For instance, the authors could compare their full model against: (1) a baseline model that does not differentiate between age groups, (2) reduced models that selectively incorporate information parameters, or (3) more complex integration models including features such as leaky integration, collapsing boundaries or multiplicative relations between ΔR and ΔI. This is important given the central role of the computational modelling in the paper.

3. Optimality and compensation – The manuscript would benefit from better defining what constitutes optimal performance in this task. Without this framework, claims about "adaptive compensation" remain speculative. The authors do not quantify the extent to which the threshold increase optimizes performance given the lower SNR. A simulation study demonstrating how different parameter combinations affect cumulative rewards would strengthen their interpretation substantially.

4. Parameter recovery – Given that the authors' conclusions rest on comparisons between model parameters, it is essential to show robust parameter recovery analysis. While they report recovery results in Suppl. Fig. 1, the details about how the recovery was performed are not specified. Additionally, it would help if the authors add the correlation between the recovered and generated parameters for each parameter. Furthermore, the authors should examine trade-offs between parameters, specifically between the drift related and response boundary parameters. This is important to establish that the results do not stem from statistical artifacts.

5. Compensation mechanism – The authors propose that older adults use higher decision thresholds to compensate for reduced SNR. However, the compensatory interpretation requires more rigorous evaluation. Does this relationship take place at the individual participant level (correlation between threshold and SNR parameters at the individual level), or is it only observable at the group level? Is this compensatory adjustment a conscious strategic adaptation or an implicit process?

Minor comments:

6. Theoretical framework – The paper proposes that differences in SNR and decision boundary between age groups, but these observations are insufficiently grounded in theoretical framework on cognitive aging and decision-making processes. Why do older adults exhibit lower SNR? Is it the result of difficulty in processing information or a matter of visual acuity? Was visual acuity controlled for?

7. Parameter distributions – The analysis would benefit from data about parameter distributions across participants rather than relying solely on group means.

8. Statistical power – While the sample size is substantial and the study appears adequately powered, the authors do not specify how sample size was determined.

9. Learning effects – Do the observed age differences remain stable throughout the task or evolve differently for older versus younger adults as they gain experience with the task structure? Comparing behavior between early and late blocks could reveal differences in how exploration strategies adapt across the experiment.

10. Figure 5 notations – The ‘i1’, ‘i2’ notations in Fig. 5 are not defined in the caption.

Reviewer #3: The manuscript "The dynamics of explore- exploit decisions suggest a threshold mechanism for reduced random exploration in older adults" by Phelps et al. describes a study that was aimed at investigating the origins of changes in explorative behaviour in older adults (65-74 years) compared to younger ones (18-30 years) using a Horizon task and drift diffusion modelling (DDM) to distinguish between directed and random exploration. Previous studies had shown effects of reduced exploration and increased exploitation in older adults. But while changes in random exploitation had been suggested as potential underlying mechanism, the exact mechanism had remained unclear.

Here, using the advantage of DDMs by fitting both choice and reaction times as part of the modelling, the authors could show lower Signal To Noise Ratio and a higher (decision) threshold in the group of older participants. They suggested that the decrease in random exploration shown in older adults might be driven by the elevated decision threshold, which could be a way of compensating for the second finding, namely a reduction in SNR.

This study asks a really interesting research question, that has become more central to the learning and decision-making literature recently with papers such as Findling et al. 2019 and 2021 discussing the different aspects of decision noise. Moreover, the study uses an appropriate task to distinguish between different aspects of explorative behaviour and employs advanced methods to address the research questions posed here. Thus, I would consider the results to be relevant for the scientific community and a valuable and interesting addition to the field of decision science. Overall, I would be in favor of this study being published, given that my concerns and questions are addressed.

Comments to the authors:

Introduction:

p. 6, ll. 93-96. Here, the sentence structure seems to be grammatically incorrect or at least it sounds confusing to me. Hence, I would suggest rewriting this sentence to make it clearer.

My next comment is regarding a specific part of the manuscript (p. 8, ll. 133-137). Unfortunately, for a reader who is not very familiar with the Horizon task, this paragraph makes a lot of assumptions and the claims seem somewhat ad-hoc. Consequently, I would suggest rewriting the paragraph and explain in a bit more detail, how they arrive at claims such as ‘This suggests that horizon-dependent directed exploration is similar in younger and older adults.’ or ‘This suggests that horizon dependent random exploration is reduced in older adults.’ that seem to be based on the findings by the study of Mizell et al.

My next concern is about the way the DDM is described in the introduction (p. 8-9, ll. 139-161). While some of what is described here might be considered common knowledge, I would still add the respective citations here. Though I realize that the authors describe the DDM extensively in the method section including the appropriate citations, given the structure of the manuscript with results coming first, this is still helpful.

Results

On p.15 it is unclear what the authors mean by spatial bias in the model, as it is not explicitly stated in this part of the paper, as far as I can see (p.15, ll. 254). I would suggest to add a short description here as well to make this easier for the reader.

p. 18 starting l. 305 the authors say ‘The degree of this reward-response time modulation appears to differ between younger and older adults, so…[…]’ This suggestion comes somewhat out of the blue, as the previous paragraph does not say so and while you probably base this on Fig 6, this is not explicitly stated and therefore confusing. It needs to be made clear, on what this statement is based on, so please consider some rewriting of this paragraph.

Methods

In general, given that neurocognitive test scores are available for both samples and the supplement describes a few key differences in cognitive scores between the groups (even though they are in the expected direction when considering age), I would be interested in how the differences between the two groups are related to or might have impacted the findings. Specifically, do the main results in terms of changes in random exploration for the older participants still hold when corrected for neurocognitive differences? Or, as one might speculate that those neurocognitive changes mediate the changes in explorative behaviour, have you assessed the mediating effect of neurocognitive functions on the association between age and exploration (SNR reduction & threshold)? I consider this especially relevant, given that the authors are listing the potential changes in crystalized intelligence in older adults as potential origin of the changes in the discussion (ll. 441-444).

I would also suggest to add some more information about the parameters in the DDM section of the paper, for instance the authors could list the 7 parameters again in brackets. This way, every reader will know right away which parameters the authors are referring to here (p. 39, ll. 663-672). Also, I was wondering, why did you choose to fix c instead of any of the other parameters?

Finally, I was wondering whether the authors could check if/ how many of the participants had chance fits when fitting the DDM and then rerun analysis without them to see whether results remain stable. This is an approach often used for reinforcement learning models, though I am not sure, whether this is a common thing done for DDMs.

General

Finally, many of the figure captions are extremely short and some of them could benefit from additional, though brief description of what is depicted in the respective panels, i.e. mini summary of the finding. While a lot of this is also described in the main text of the manuscript, figures and figure captions need to be treated as ‘stand-alone’ items and in theory should be comprehensible without the main manuscript. Also, some information is simply missing from the captions, such as in Fig 5. Here, it is unclear what i1-i4 means and the reader has to search the manuscript for this information.

**Have the authors made all data and (if applicable) computational code underlying the findings in their manuscript fully available?**

Reviewer #1: Yes

Reviewer #2: **No: **

Reviewer #3: Yes

PLOS authors have the option to publish the peer review history of their article (what does this mean? ). If published, this will include your full peer review and any attached files.

**Do you want your identity to be public for this peer review?** For information about this choice, including consent withdrawal, please see our Privacy Policy .

Reviewer #1: No

Reviewer #2: No

Reviewer #3: No

**Figure resubmission:**

**Reproducibility:**



---

## [Decision Letter · Decision Letter 1]

2 Sep 2025

PCOMPBIOL-D-25-00273R1

The dynamics of explore-exploit decisions suggest a threshold mechanism for reduced random exploration in older adults

PLOS Computational Biology

Dear Dr. Phelps,

Thank you for submitting your manuscript to PLOS Computational Biology. After careful consideration, we feel that it has merit but does not fully meet PLOS Computational Biology's publication criteria as it currently stands. Therefore, we invite you to submit a revised version of the manuscript that addresses the points raised during the review process.

The three reviewers agree on the quality of your response to their comments. However, two of them still express concerns regarding your model selection and comparisons. We agree that these points should be accounted for. 

Please submit your revised manuscript within 60 days Nov 02 2025 11:59PM. If you will need more time than this to complete your revisions, please reply to this message or contact the journal office at ploscompbiol@plos.org. Please include the following items when submitting your revised manuscript:

We look forward to receiving your revised manuscript.

Kind regards,

Hugues Berry

Section Editor

PLOS Computational Biology

**Journal Requirements:**

**Reviewers' comments:**

Reviewer's Responses to Questions

Reviewer #1: The authors have done a good job addressing my comments.

Reviewer #2: I thank the authors for addressing my comments.

My main remaining concern is the model comparison (Fig. S6). The results indicate that a model yoking H1 & H6 parameters achieves the lowest BIC (preferred for ~70% of participants), whereas the flexible baseline performs worst (preferred for fewer than 10%). Because DDM parameters are model dependent, parameter differences derived from the statistically inferior baseline model raise overfitting concerns and are not justified as presented.

I believe that instead of burying this analysis in the supplement, the authors should be more transparent about the results, move them to the main text, and reframe and soften claims relying on the flexible specification. Approaches I consider preferable are: 1. Compute parameters via model averaging (based on BIC or AIC weights) and use those parameters for group/horizon comparisons. 2. Compare models using cross-validation and show that the gain (in likelihood terms) for the flexible model is substantial. 3. Model the task differently, for example, a model assuming a starting-point bias might fit better than one assuming a change in drift.

I leave it to the authors and the editor to decide which approach to take.

Reviewer #3: First of all, all of my comments have been addressed by the authors and the appropriate revisions have been made to the manuscript.

However, I am concerned after having also gone over the issue raised by the other two reviewers regarding the potential need of extending the model space by additional models and the authors‘ response to this. I agree with the concern raised here, even though I had not raised it myself.

I personally find the argument made by the authors of why they eventually chose to stick to the baseline (original model) somewhat arbitrary. They say that they would want to stick to the original model given that quite a number of participants in each sample are best fit by this model or one of the two new models including a yoked threshold or SNR reward and that they want to retain flexibility for those participants.

While it is true that about 55 participants are fit best by one of the other models, with the least number of people fit best using the original baseline model, a very large part of the sample is fit better by the fourth model (from what I can tell based of the figure, more than 80 and 100 participants respectively), and this applies to both the older and younger sample. It does not really make sense to use model comparison if one just picks the model that one originally intended to analyze, independent of what the BIC indicates. Also, even when sticking to the original model and ignoring the results of model comparison, these results just seem to be included in the supplement without any further discussion of why so many participants might have been better fit by another model or what the implication might be for the results. I feel like this is not sufficient here and would suggest additional analysis including the winning model or at least a further discussion of the modelling results including the discrepancy of the analyzed model vs. the winning model.

**Have the authors made all data and (if applicable) computational code underlying the findings in their manuscript fully available?**

Reviewer #1: Yes

Reviewer #2: None

Reviewer #3: Yes

PLOS authors have the option to publish the peer review history of their article (what does this mean? ). If published, this will include your full peer review and any attached files.

**Do you want your identity to be public for this peer review?** For information about this choice, including consent withdrawal, please see our Privacy Policy .

Reviewer #1: No

Reviewer #2: No

Reviewer #3: No

**Figure resubmission:**

**Reproducibility:**



---

## [Decision Letter · Decision Letter 2]

2 Dec 2025

Dear Dr Phelps,

We are pleased to inform you that your manuscript 'The dynamics of explore-exploit decisions suggest a threshold mechanism for reduced random exploration in older adults' has been provisionally accepted for publication in PLOS Computational Biology.

Best regards,

Hugues Berry

Section Editor

PLOS Computational Biology

Hugues Berry

Section Editor

PLOS Computational Biology

Reviewer's Responses to Questions

**Comments to the Authors:**

Reviewer #2: The authors have addressed all my comments. I believe leave-one-out cross-validation is a more appropriate metric than BIC for the context of the paper.

Reviewer #3: The authors have adequately addressed my earlier concerns regarding the model comparison and model selection. I have no further comments at this point.

**Have the authors made all data and (if applicable) computational code underlying the findings in their manuscript fully available?**

Reviewer #2: Yes

Reviewer #3: Yes

PLOS authors have the option to publish the peer review history of their article (what does this mean? ). If published, this will include your full peer review and any attached files.

**Do you want your identity to be public for this peer review?** For information about this choice, including consent withdrawal, please see our Privacy Policy .

Reviewer #2: No

Reviewer #3: No

---

## [Editor Report · Acceptance letter]

PCOMPBIOL-D-25-00273R2

The dynamics of explore-exploit decisions suggest a threshold mechanism for reduced random exploration in older adults

Dear Dr Phelps,

I am pleased to inform you that your manuscript has been formally accepted for publication in PLOS Computational Biology. Your manuscript is now with our production department and you will be notified of the publication date in due course.

With kind regards,

Anita Estes
